# Application of Cortical Bone Plate Allografts Combined with Less Invasive Stabilization System (LISS) Plates in Fixation of Comminuted Distal Femur Fractures

**DOI:** 10.3390/medicina59020207

**Published:** 2023-01-20

**Authors:** Zhimin Guo, Hui Liu, Deqing Luo, Taoyi Cai, Jinhui Zhang, Jin Wu

**Affiliations:** Department of Orthopaedics, The 909th Hospital, School of Medicine, Xiamen University, Zhangzhou 363000, China

**Keywords:** distal femur fracture, cortical bone plate allografts, LISS plates

## Abstract

*Background and Objectives:* At present, the management of comminuted distal femur fractures remains challenging for orthopedic surgeons. The aim of this study is to report a surgical treatment for comminuted distal femur fractures using supplementary medial cortical bone plate allografts in conjunction with the lateral less invasive stabilization system (LISS) plates. *Materials and Methods:* From January 2009 to January 2014, the records of thirty-three patients who underwent supplementary medial cortical bone plate allografts combined with lateral LISS plates fixation were reviewed. Clinical and radiographic data were collected during regular postoperative follow-up visits. Functional outcomes were determined according to the special surgery knee rating scale (HSS) used at the hospital. *Results:* Thirty patients were followed for 13 to 73 months after surgery, with an average follow-up time of 31.3 months. The mean time to bone union was 5.4 months (range of 3–12 months) and the mean range of knee flexion was 105.6° (range of 80–130°). Of the remaining patients, 10 had a score of “Excellent”, while 10 had a score of “Good”. Three patients had superficial or deep infections, one patient had nonunion that required bone grafting, and one patient had post-traumatic knee arthritis. *Conclusions:* Based on these promising results, we propose that supplementary medial cortical bone plate allografts combined with lateral LISS plate fixation may be a good treatment option for comminuted distal femur fractures. This treatment choice not only resulted in markedly improved stability on the medial side of the femur, but also satisfactory outcomes for distal femoral fractures.

## 1. Introduction

Distal femoral fractures comprise approximately 3–6% of all femoral fractures [1]. Up to now, effective treatment of comminuted distal femur fractures remains difficult for orthopedic surgeons. These fractures are often unstable and comminuted, typically resulting either from falls in female patients older than 75 years or as a result of the high-energy activities common amongst adolescent boys and men aged 15 to 24 years [2]. Classification of distal femur fractures was first described by Müller et al. and expanded in the AO/OTA classification [3,4]. These classifications are based on fracture location and pattern and are useful in determining treatment and prognosis. With the development of improved internal fixation devices, operative treatment can now produce better results than nonoperative treatment. This is especially true for comminuted supracondylar and intercondylar femur fractures [5].

A complete set of instruments and familiarity with their use are required for surgical treatment of comminuted distal femur fractures. Condylar buttress plates, dynamic condylar screws (DCS), intramedullary nailing, LISS plates, and external fixation were introduced to facilitate the treatment of these types of fractures [6,7,8,9]. However, the spectrum of injuries is so great that no single implant has been found to be suitable for every case. Moreover, patient outcomes with these types of fractures are generally unsatisfactory due to the proximity of the fracture to the knee joint [10], meaning that regaining full knee motion and function may be difficult, and significant complications such as malunion, nonunion, infection, malrotation, and implant failure occur at relatively high rates in many reports [11,12,13,14,15].

Given these challenges, this study investigated a surgical treatment strategy for comminuted distal femur fractures. The fractures included in this study were in accordance with AO/OTA classification, consisting of patients with either type A3 fractures involving distal shaft comminution, type C2 fractures involving metaphyseal comminution, or type C3 fractures characterized by metaphyseal and intra-articular comminution. The approach described here features the use of a supplementary medial cortical bone plate allograft in conjunction with a lateral LISS plate. Therapeutic effects were assessed in patients, with an average follow-up time of 31.3 months.

## 2. Patient and Method

### 2.1. Clinical Data

This study was a retrospective analysis of existing clinical cases and was approved by the institutional review board. Written informed consent was obtained preoperatively for all patients. Thirty-three patients (twenty males and thirteen females) were enrolled in the study between January 2009 and January 2014. All patients were diagnosed according to clinical presentation, X-ray, and computer tomography (CT) scans. Study participants were evaluated postoperatively every 1–2 months in the outpatient clinic.

### 2.2. Preoperative Preparation

Proximal tibial skeletal traction was performed immediately after all patients with closed fractures were admitted to the hospital. Patients with Gustilo I and Gustilo II open fractures first underwent debridement and suturing, after which they received proximal tibial skeletal traction. In two patients with a Gustilo III fracture, limited internal fixation combined with external fixation was implemented following debridement. All patients with open fractures received postoperative intravenous antibiotics for 24 to 48 h. X-ray and CT examinations were used to visualize fracture displacement and the presence of fragments when determining the surgical strategy for each patient. All patients underwent surgical treatment as soon as their condition had stabilized.

### 2.3. Surgical Procedure

Prophylactic antibiotics were given 30 min prior to surgery. No tourniquet was used. The patient was placed under either general or spinal anesthesia and then positioned in a supine position with a bolster under the knee to acquire 20–30° of flexion. This was performed in order to relax the deforming force of the gastrocnemius. For type A3 fractures, a 4–5 cm lateral incision was made just proximal to the joint line. A distal femoral LISS plate (AO, Synthes Inc., West Chester, PA, USA) was slipped under the vastus lateralis proximally and provisionally fixed distally using K wires. Close reduction was accomplished using traction and external manipulation and confirmed under fluoroscopy. During the procedure, specific attention was paid to limb alignment and length. When the position of the LISS plate was deemed satisfactory, three to six locking screws were inserted in the distal and proximal part of the bone, respectively.

For type C2 and C3 fractures, an incision was made on the lateral condyle of the femur and elongated to the tibial tubercle to fully expose the anterior and lateral aspects of the femoral condyle. Intercondylar fractures were then reduced and fixed with cannulated screws (AO) to form the supracondylar fracture. These fractures were then treated as type A3 fractures. For patients with implant failure after surgery, the lateral parapatellar approach was used to remove the implant. After the fracture was fully exposed, any scar tissue and sclerotic bone was excised, and the medullary cavity was reamed. An appropriate length LISS plate was then used to fix the fracture, and autologous iliac bone was implanted.

A suitable width cortical bone plate allograft (Xin Kang Chen Medical Technology Development Co., Ltd., Beijing, China) was selected and trimmed with a wire saw. The sharp edge of the cortical bone plate allograft was filed with a bone file, and the tip was rounded and obtuse. A 4–5 cm anteromedial incision was made along the anterior margin of the pes anserinus, following the adductor canal. The fascial envelope surrounding the vastus medialis along the posterior margin of the muscle was then incised. Blunt dissection was used to elevate the muscle off the periosteum and the intermuscular septum from the adductor tubercle to the intact proximal femoral shaft. Next, a periosteal elevator was used to strip the region between the periosteum and adductor muscles of the thigh. The prepared cortical bone plate allograft was implanted via the anteromedial incision and placed on the opposite side of the LISS plate. The LISS plate and cortical bone plate allograft were fixed in place with cortical bone screws. At least two screws were used at the distal and proximal ends of the bone plate. Finally, the open wound was rinsed and the incisions were closed, with a suction drain at the surgical site.

### 2.4. Postoperative Management

All patients received postoperative intravenous antibiotics for 24 h. Suction drains were removed on day 2–3. Active and passive range-of-motion exercises were then started. Full weight-bearing activity was allowed after a bridging callus was observed on radiographs.

### 2.5. Outcome Assessment

Outcomes after surgery were evaluated according to HSS scores, which rely on a 100-point scoring system that assesses pain (30 points), function (22 points), range of motion (18 points), muscle strength (10 points), flexion deformity (10 points), and joint stability (10 points). Overall, “Excellent” was classified as a cumulative score of 85 or more, “Good” as 70 to 84, “Fair” as 60 to 69, and “Poor” as 60 or less. Postoperative functional results were obtained regularly. Postoperative radiological parameters, including X-rays and CT scans, were taken every four weeks to evaluate bony fusion.

## 3. Results

Detailed clinical patient parameters are shown in Table 1. The average age at enrollment was 44.5 years (range was 18–78 years). Follow-up visits were conducted with thirty patients between 13 and 73 months post-operation, with an average follow-up time of 31.3 months. One patient stopped responding after a 3-month follow-up visit, and two patients lost connection at the 6-month follow-up visit. Twenty-nine patients suffered from closed fractures and four had open fractures (1 Gustilo I, 1 Gustilo II, 1 Gustilo IIIA, and 1 Gustilo IIIB). According to the AO/ASIF system, 33 fractures were classified as the following: A3 (n = 10), C2 (n = 13), and C3 (n = 10). The causes of injury included traffic accidents (20 patients, 60.6%), heavy object crush injuries (5 patients, 15.1%), falls from a significant height (6 patients, 18.2%), and implant failure (2 patients, 6.1%). Eight patients presented with complicated injury. Two patients had fractures associated with an ipsilateral tibial fracture (including one popliteal artery injury patient), two with a hemopneumothorax, two with a traumatic brain injury, two with contralateral tibial and fibula fractures, and one with an ipsilateral patella fracture. Due to the severity of associated hemopneumothorax and traumatic brain injury, neurosurgical or thoracic treatments were performed on patients before attending the lower limb fractures [16].

The mean time to bone union (formation of a circumferential bridging callus across the fracture) was 5.4 months (range was 3–12 months). Three patients stopped visits and ceased communication during the follow-up period (Patients 21, 28, and 33). Outcomes for the remaining patients were “Excellent” for 10 and “Good” for 10, making the percentage of combined “Excellent” and “Good” scores 67.7%. The mean range of knee flexion was 105.6° (range of 80–130°). More specifically, 2 patients had an 80° range, 4 patients had a 90° range, 7 had a 100° range, 12 had a 110° range, and 5 had a ≥120° range of knee flexion. All patients achieved full knee extension. Three patients had weakness in their quadriceps, but all others attained full quadricep strength. Six patients had the implant removed (Table 2).

One patient had a deep infection five days after the operation and underwent a secondary surgery (implant removal and external fixation). There were two patients who had minor surgical complications, including one superficial wound infection and one partial wound dehiscence. After debridement and suturing, both patients’ complications were resolved. One patient with nonunion required bone grafting without hardware exchange. Post-traumatic arthritis was seen in one patient at the final follow-up, which was based on the radiologic assessment (Table 2). Typical cases are shown in Figure 1 (Patient 2), Figure 2 (Patient 5), and Figure 3 (Patient 10).

## 4. Discussion

Comminuted distal femur fractures are frequently associated with severe comminution, substantial soft tissue injury, and bone defects. Prior to the 1970s, nonoperative management was the treatment of choice [2]. With the steady improvement of surgical techniques and implants, operative fixation has gained widespread acceptance. Historically, these fractures were treated with condylar buttress plates [6]. Gradually, retrograde nails and DCS took the place of condylar buttress plates. This shift was due to their superior biomechanical design that resulted in decreased varus collapse events when compared with the results using standard condylar buttress plates [17]. The indication of DCS is noncomminuted periarticular fractures without coronal splits and with good bone quality [18]. Recently, locking plates have become the main treatment for comminuted distal femur fractures, particularly for supracondylar and intercondylar comminuted femur fractures. With the increased number of fixation screws used in the distal femur metaphysis, locking plates provide increased biomechanical resistance and stability [19]. However, perioperative and postoperative complications such as malunion, nonunion, implant failure, malrotation, and infection are still common with this approach [14,15].

The main reasons for implant failure are primarily due to the following problems: (1) high bending stress exerted on the laterally placed plates in the presence of marked cortical defects and (2) locking plates are usually implanted using the minimally invasive percutaneous plate osteosynthesis (MIPPO) technique. Since the MIPPO technique is relatively short range and intraoperative fluoroscopy has a limited range, there is a high incidence (approximately 30%) of axial malalignment after surgery. Axial malalignment results in increased load on the plate, which can cause implant failure. Here, implant failure was found in two patients over the age of 60 who had been initially treated with a single-side plate and screws, followed by additional operations as needed. The current treatment approach for implant failure features scar tissue removal and large amounts of autologous iliac bone grafts, as well as implant replacement. Bilateral autologous iliac bone grafts have often been applied to repair cortical defects, which can increase surgical trauma and the chance of infection. Furthermore, the stability immediately following the structural allograft cannot support early postoperative functional exercise, which is important for recovery. Therefore, we performed a medial implant of the cortical bone plate allograft integrated with a lateral LISS plate for the two patients with implant failures. Patient 2 was a 61-year-old male patient with a type A3 fracture that had been initially treated using dynamic condylar screws. The implant failure was observed seven months following surgery and required reoperation. After treatment with a cortical bone plate allograft combined with LISS plate fixation, bone union was observed five months later (Figure 1).

Types C2, C3, and partial A3 fractures of the distal femoral are prone to induce nonunion and implant failure, particularly in the cases of severe cortical defects in the medial femur. On the basis of lateral LISS plate implantation using MIPPO technology, a suitable length and width allogeneic cortical bone plate was implanted from the medial epicondyle of the femur, which achieved an integrated fixation of the triangular support and avoided excessive elevation of the periosteum at the fracture site. For severe comminuted fractures and/or periprosthetic fractures of the distal femur, double plating with autogenous bone grafting executed via a modified Olerud extensile approach was also used. Although acceptable clinical outcomes were achieved, there are some limitations to this approach, including excessive elevation of the periosteum, large trauma (tibial tuberosity osteotomy), and lack of integrated fixation [20]. Recently, a double-plating technique was used for the treatment of supracondylar femur fractures. Based on promising follow-up results, they recommended this technique specifically for patients with poor bone quality, comminuted fractures, and very low periprosthetic fractures [21]. However, no detailed functional outcomes were described in their results, and some important points needed to be clarified [22].

The application of allogeneic cortical bone plates in repairing bone defects has been frequently reported, and satisfactory clinical results have been achieved with this approach [23,24]. Moreover, allogeneic cortical bone plates were used in the treatment of periprosthetic fractures of the femur [25] and distal femoral nonunion [26]. However, there are not many relevant reports about the use of allogeneic cortical bone plates in the treatment of comminuted fractures of the distal femur. In our study, an allogeneic cortical bone plate was used in the treatment of comminuted distal femur fractures, which has the following advantages: (1) wide scope of application; (2) the union of the allogeneic cortical bone plate and host bone can reconstruct cortical defects of the medial femur, and when combined with an autologous iliac bone graft, this treatment has a strong osteoinductive effect and can promote bone healing; (3) LISS plates and allogeneic cortical bone plates were implanted using MIPPO technology, which minimized periosteal elevation and disruption of blood supply at the fracture site, which not only increased fixation rigidity, but also contributed to fracture healing; (4) allogeneic cortical bone plates are a biomechanically sound alternative to metal plates fixed with screws, and could markedly improve stability and rigidity after lateral LISS plate fixation; (5) utilization of an LISS plate and allogeneic cortical bone plate presented firm integrated fixation of the triangular support. Furthermore, knee function exercises were conducted soon after the operation, resulting in overall better therapeutic outcomes.

Taken together, we believed that a cortical bone plate allograft combined with the LISS plate fixation technique may be an option for the treatment of comminuted distal femur fractures and is beneficial for early weight-bearing following surgery. However, there are some limitations to this study, including its retrospective nature with old data, the relatively small group of patients studied, and a combining of young and old populations. A long-term RCT study with a larger number of patients and control groups that include other fixation methods should be performed to further validate our findings here.

## 5. Conclusions

Biomechanical and clinical studies suggested early weight-bearing may be performed immediately following surgical treatment of comminuted distal femur fractures, that fixation failure was associated with medial comminution, and that medial comminution should be managed with additional fixation [27]. Thus, we recommend that a cortical bone plate allograft combined with the LISS plate fixation technique be used for treatment of comminuted distal femur fractures, especially indicated in cases of severe medial femur cortical defects and implant failure after surgery.

## Figures and Tables

**Figure 1 medicina-59-00207-f001:**
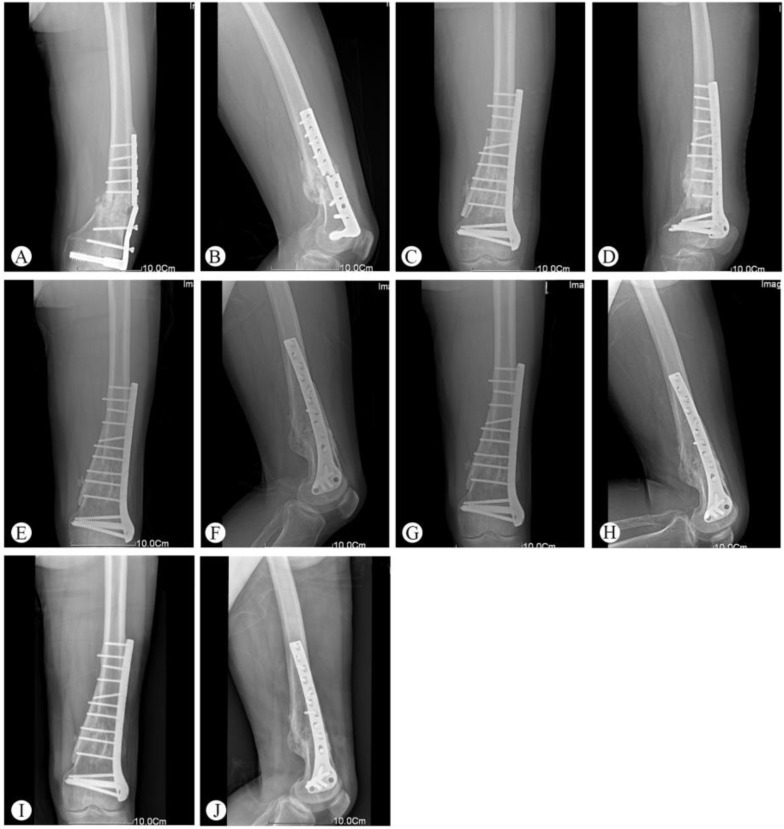
Representative images of Patient 2 (61-year-old male patient with a type A3 fracture that had been initially treated with dynamic condylar screws). (**A**,**B**) Implant breakage was observed seven months after surgery. (**C**,**D**) X-ray at 5 months after operation. Bone union was observed. (**E**,**F**) Follow-up X-ray at 36 months. (**G**,**H**) Follow-up X-ray at 60 months. (**I**,**J**) Follow-up X-ray at 73 months.

**Figure 2 medicina-59-00207-f002:**
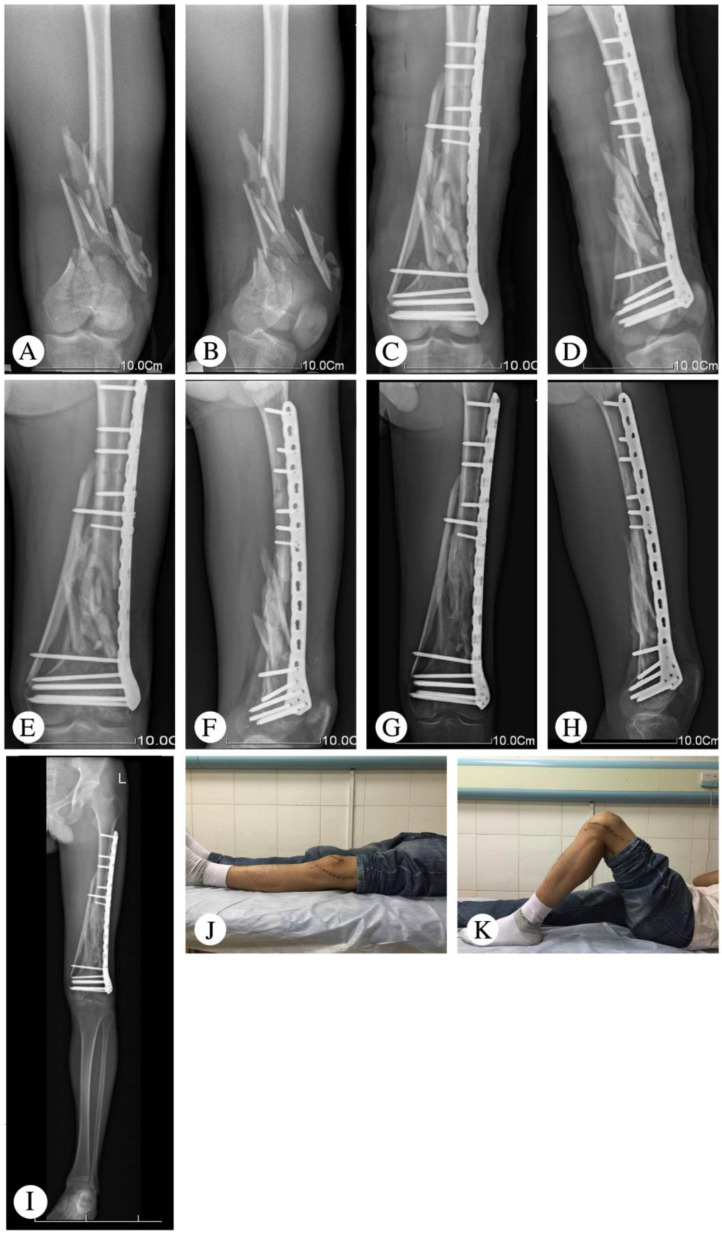
Representative images of Patient 5 (29-year-old male who suffered a heavy object crush to his left thigh). (**A**,**B**) X-ray at admission. (**C**,**D**) X-ray 5 days after the operation. (**E**,**F**) Follow-up X-ray at 3 months. (**G**,**H**) Follow-up X-ray at 30 months. (**I**) Full-length radiography showing the lower limb at a 33-month follow-up visit. Limb alignment and length was good. (**J**,**K**) Range of knee joint motion at a 33-month follow-up visit. The patient achieved full knee extension. However, the range of knee flexion was only 90°.

**Figure 3 medicina-59-00207-f003:**
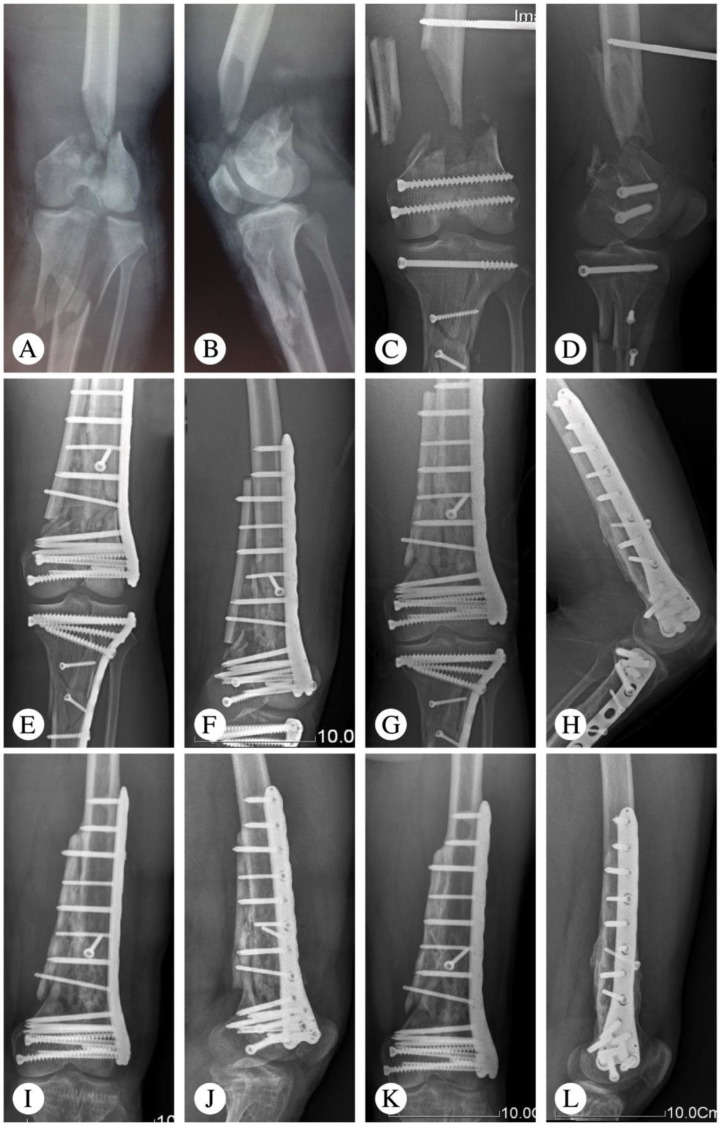
Representative images of Patient 10 (21-year-old male patient with a Gustilo IIIB fracture resulting from a traffic accident). (**A**,**B**) X-ray at admission. (**C**,**D**) X-ray after emergency operation. Limited internal fixation combined with external fixation was utilized. (**E**,**F**) X-ray 3 months after an interfixation operation. (**G**,**H**) Follow-up X-ray at 9 months. (**I**,**J**) Follow-up X-ray at 12 months. Removal of the tibial implant. (**K**,**L**) Follow-up X-ray at 69 months.

**Table 1 medicina-59-00207-t001:** Clinical parameters of the patients.

Patients No	Gender	Age (Years)	Causes of Injury	Injury Type	Fracture Type	Other Injury
1	Male	40	Heavy object crushes	Closed fracture	C2	-
2	Male	61	Implant failure	Closed fracture	A3	-
3	Male	18	Fall from height	Closed fracture	A3	-
4	Male	35	Traffic accident	Closed fracture	C2	-
5	Male	29	Heavy object crushes	Closed fracture	C3	-
6	Female	69	Fall from height	Closed fracture	C2	-
7	Female	40	Traffic accident	Closed fracture	C3	-
8	Male	41	Fall from height	Closed fracture	C2	Traumatic brain injury
9	Female	22	Traffic accident	Closed fracture	C2	-
10	Male	21	Traffic accident	Open fracture	C3, Gustilo III b	Ipsilateral tibial fracture
11	Male	30	Traffic accident	Closed fracture	C2	Contralateral tibial and fibula fracture
12	Female	34	Traffic accident	Open fracture	A3, Gustilo I	-
13	Male	23	Heavy object crushes	Closed fracture	A3	-
14	Female	69	Implant failure	Closed fracture	C3	-
15	Female	31	Traffic accident	Closed fracture	C2	-
16	Male	40	Traffic accident	Closed fracture	C2	Ipsilateral tibial fracture
17	Male	23	Traffic accident	Closed fracture	A3	-
18	Female	59	Traffic accident	Closed fracture	C3	Hemopneumothorax
19	Male	55	Traffic accident	Closed fracture	C3	-
20	Female	71	Fall from height	Closed fracture	A3	-
21	Female	47	Heavy object crushes	Closed fracture	C2	Ipsilateral patella fracture
22	Male	33	Traffic accident	Open fracture	A3, Gustilo III a	-
23	Male	42	Traffic accident	Closed fracture	C2	Traumatic brain injury
24	Male	51	Traffic accident	Closed fracture	C3	-
25	Female	56	Traffic accident	Closed fracture	C2	Contralateral tibial and fibula fracture
26	Male	59	Traffic accident	Closed fracture	A3	-
27	Female	67	Fall from height	Closed fracture	C3	-
28	Female	42	Heavy object crushes	Closed fracture	C2	Hemopneumothorax
29	Male	53	Traffic accident	Open fracture	A3, Gustilo II	-
30	Male	46	Traffic accident	Closed fracture	C3	Contralateral tibial and fibula fracture
31	Male	33	Traffic accident	Closed fracture	C2	-
32	Male	78	Traffic accident	Closed fracture	A3	-
33	Female	51	Fall from height	Closed fracture	C3	-

**Table 2 medicina-59-00207-t002:** Outcomes of the patients.

Patients No	Follow-Up (Months)	Bone Union (Months)	Knee Range of Motion	Outcomes *	Complications
1	14	4	110°	Good	-
2	73	5	100°	Fair	Quadricep strength grade 3
3	14	8	80°	Poor	Deep infection, secondary surgery
4	26	5	100°	Excellent	-
5	33	5	90°	Fair	Quadricep strength grade 4
6	29	6	110°	Excellent	-
7	28	7	100°	Good	-
8	35	6	110°	Excellent	-
9	22	3	110°	Good	-
10	69	9	80°	Poor	Quadricep strength grade 4
11	19	3	130°	Excellent	-
12	28	5	110°	Good	-
13	15	4	100°	Good	-
14	17	6	90°	Fair	-
15	26	7	110°	Fair	Superficial infection
16	31	5	120°	Excellent	-
17	43	4	130°	Excellent	-
18	19	5	100°	Fair	-
19	54	12	90°	Fair	Nonunion, secondary surgery
20	27	3	110°	Excellent	-
21	Lost to follow-up	-	-	-	-
22	50	8	110°	Good	-
23	25	5	120°	Excellent	-
24	60	4	110°	Good	Post-traumatic arthritis
25	24	5	110°	Good	-
26	17	4	110°	Excellent	-
27	42	6	110°	Good	-
28	Lost to follow-up	-	-	-	-
29	45	4	90°	Good	Superficial infection
30	26	4	100°	Fair	-
31	13	4	100°	Fair	-
32	33	5	120°	Excellent	-
33	Lost to follow-up	-	-	-	-

* Based on knee rating scale of the Hospital for Special Surgery.

## Data Availability

The analyzed datasets of this study are available from the corresponding author on reasonable request.

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
