# Peer review of "Application of Cortical Bone Plate Allografts Combined with Less Invasive Stabilization System (LISS) Plates in Fixation of Comminuted Distal Femur Fractures"

_medicina, 2023, doi:10.3390/medicina59020207_

Round 1
Reviewer 1 Report
This is an interesting case series with comminuted distal femur fractures.
However there are some aspects that require your attention:
1. You present 33 patients, but afterwards 3 of them are lost in the follow up, you could remove those 3 cases from the entire study.
2. The case series seems old, the approval of the ethics board is from 2008. Please explain in the discussions section of the manuscript why it took so long to gather this group of patients from 2009-2014.
3. Moreover you end the enrollment of the patients in 2014 and the follow up period is of maximum 73 months. Please specify if the last patients was enrolled in 2014 and you monitor the patient for 6 years, obviously you had all the data collected till 2020. Did you postpone publishing the data due to the Covid-19 pandemics? Why did you stop enrolling cases between in 2014, because a greater number of cases could have promoted your study from a case series to a cohort study.
4. Explain if LISS is a new technique, using a new device, or a standard of practice, but with a lower incidence and prevalence of cases benefiting from this technique?
5. You mention evaluating the functional outcomes with special surgery knee rating scale (HSS) used at the hospital. Please in the Materials and Method Section of the manuscript describe this rating scale, because it seems to be an instrument designed in your local hospital.
6. Some of the cases presented with head trauma. Expand on this subject, because probably the severity of this trauma implied performing other neurosurgical or ENT or OMF surgeries on the patients before attending the lower limb fractures. Reference this to newer articles such as Dumitru M, Vrinceanu D, Banica B, Cergan R, Taciuc IA, Manole F, Popa-Cherecheanu M. Management of Aesthetic and Functional Deficits in Frontal Bone Trauma. Medicina (Kaunas). 2022 Nov 30;58(12):1756. doi: 10.3390/medicina58121756. PMID: 36556958; PMCID: PMC9781007.
7. At the Author Contribution section please format it according to MDPI requirements.
Wish you a New Year 2023 and hope to see your improved manuscript soon.
Author Response
Reviewer #1: This is an interesting case series with comminuted distal femur fractures. However there are some aspects that require your attention:
- You present 33 patients, but afterwards 3 of them are lost , you could remove those 3 cases from the entire study.
Thanks for your constructive suggestion. First of all, this study was a retrospective analysis of existing clinical cases. Secondly, Table I showed the clinical parameters of the 33 patients, including gender, age, causes of injury, injury type, fracture type and complicated injury. Thus, restoration of the 3 cases lost in the follow up presents the characteristics of the series cases to readers better.
- The case series seems old, the approval of the ethics board is from 2008. Please explain in the discussions section of the manuscript why it took so long to gather this group of patients from 2009-2014.
Sorry for the confusing. We spent 5 years to gather this group of 33 patients due to the respect for patients informed consent right. Some suitable patients refused to use cortical bone plate allografts and chose to use double plate.
- Moreover you end the enrollment of the patients in 2014 and the follow up period is of maximum 73 months. Please specify if the last patients was enrolled in 2014 and you monitor the patient for 6 years, obviously you had all the data collected till 2020. Did you postpone publishing the data due to the Covid-19 pandemics? Why did you stop enrolling cases between in 2014, because a greater number of cases could have promoted your study from a case series to a cohort study.
Thank you very much for your comprehension. Indeed, we expected to publish our article at the beginning of 2020. Due to the Covid-19 pandemics in China, we had to postpone publishing the data. The cortical bone plate allograft used in the series cases was provided by Xin Kang Chen Medical Technology Development Co., Ltd, Beijing, China. Unfortunately, the purchase contract between our hospital and the Xin Kang Chen Medical Technology Development Co., Ltd expired in 2015, and there was no renewal of the contract. Therefore, we had to stop enrolling cases in 2014.
- Explain if LISS is a new technique, using a new device, or a standard of practice, but with a lower incidence and prevalence of cases benefiting from this technique?
Sorry for the confusing. Less Invasive Stabilization System (LISS) is a new type of internal fixation system based on the principle of minimally invasive surgery and the advantages of interlocking intramedullary nail technology and biological bone bonding technology. In 1990, Association for the Study of Internal Fixation (AO/ASIF) developed a new type of internal fixation product-minimally invasive fixation system (LISS). The internal fixation system is suitable for the fixation of comminuted fractures of distal femur and proximal tibia, especially for osteoporotic patients and periprosthetic fractures. Nowadays, LISS is not only a internal fixation system, but also concept of the minimally invasive surgery and has become a standard of practice for the treatment of comminuted fractures of distal femur and proximal tibia.
References:
- Weight M, Collinge C. Early results of the less invasive stabilization system for mechanically unstable fractures of the distal femur (AO/OTA types A2, A3, C2, and C3). J Orthop Trauma. 2004;18(8):503-50
- Rodríguez-Roiz JM, Seijas R, Camacho-Carrasco P, Zumbado JA, Sallent A, Ares-Rodríguez O. LISS plate for treatment of distal femur fracture. Clinical and functional outcomes. Acta Orthop Belg. 2018;84(3):316-320.
- You mention evaluating the functional outcomes with special surgery knee rating scale (HSS) used at the hospital. Please in the Materials and Method Section of the manuscript describe this rating scale, because it seems to be an instrument designed in your local hospital.
Sorry for the confusing. The HSS knee score (HSS) is a scoring system proposed in 1976 by the American Second Hospital of Special Surgery to evaluate knee joints before and after surgical procedures. This scale assesses six key aspects: pain, function, ROM, muscle strength, knee flexion deformity, and knee instability. The total possible HSS score is 100 points (pain: 30 points; function: 22 points; Range of motion: 18 points; muscle strength: 10 points; knee flexion deformity: 10 points; knee instability: 10 points). The results are divided into four grades: excellent (> 85 points), good (70–84 points), moderate (60–69 points), and poor (< 59 points). Now, it has been widely used in the postoperative functional evaluation of peri-knee fractures, including distal femur fractures. Moreover, we have add the description of this rating scale in the Patient and Method Section of the manuscript.
References:
- Gurkan V, Orhun H, Doganay M, SalioÄŸlu F, Ercan T, Dursun M, Bülbül M. Retrograde intramedullary interlocking nailing in fractures of the distal femur. Acta Orthop Traumatol Turc. 2009;43(3):199-205.—According to the modified HSS knee scale, the results were excellent in five femurs (29.4%), good in six femurs (35.3%), moderate in five femurs, and poor in one femur (5.9%).
- Liu F, Tao R, Cao Y, Wang Y, Zhou Z, Wang H, Gu Y. The role of LISS (less invasive stabilisation system) in the treatment of peri-knee fractures. Injury. 2009 Nov;40(11):1187-94.—Functional assessment was performed using HSS (hospital for special surgery) score.
- Some of the cases presented with head trauma. Expand on this subject, because probably the severity of this trauma implied performing other neurosurgical or ENT or OMF surgeries on the patients before attending the lower limb fractures. Reference this to newer articles such as Dumitru M, Vrinceanu D, Banica B, Cergan R, Taciuc IA, Manole F, Popa-Cherecheanu M. Management of Aesthetic and Functional Deficits in Frontal Bone Trauma. Medicina (Kaunas). 2022 Nov 30;58(12):1756. doi: 10.3390/medicina58121756. PMID: 36556958; PMCID: PMC9781007.
Thanks for your constructive suggestion. We have read the newer articles you recommended carefully (For example Management of Aesthetic and Functional Deficits in Frontal Bone Trauma.) and add the relevant content in the Patient and Method Section of the manuscript.
- At the Author Contribution section please format it according to MDPI requirements.
We have corrected it according to MDPI requirements and please see the revised manuscript.

Reviewer 2 Report
Generally this is an older technique. Surgeons have been using allograft struts for decades. This is never acknowledged by the authors. It should be.
Many things about this paper need rewriting. See below. Why did it take 14 years to publish as the technology has improved a lot over the last 5 years? Why combine the older patients with the young as the young do not generally need medial fixation like the older patients. Also, older patients need to be full wt bearing immediately and not non wt bearing as mentioned. This needs to be rewritten along 2023 standards.
Introduction - Line 42-43 - This sentence is not needed - adds nothing.
- Line 50 - Full WB now standard for elderly patients after fixation of these fractures. Please change this writing as old ideas do not make for good publishing.
- Line 53 - This is not a novel idea as allografts have been used with cerclage wires for peri-prosthetic fractures for decades. Please acknowledge and change this paragraph.
Patients and Methods - Why lump young and old patients together? They are different situations entirely. This reviewer recommends eliminating the few older patients and keeping the young patients (less than 60yo) and making this a high energy paper. Most young people do not need a graft to heal with good technique. How do you justify using it in this young population?
Line 64-83 - Most of this writing is "Results" - move to the next section. Why do you have 14 year old data? Why publishing so late when it is out of date? Much has changed over the last 14 years.
Line 120 - Was there an incision proximally too? Please explain how screws were implanted proximally.
Line 129 - Were the LISS and the allograft connected by fixation or independent?
Discussion - Line 231 - You talk of periprosthetic fractures - were there some in your patient list?
The Strengths and Limitations are mentioned very superficially in the Conclusions. Move to their own paragraph in the Discussion. Make more clear as there are problems such as 1) old data - 14 years old and much has changed in this time period. 2) Combining young and old population, 3) weightbearing was nonWB when it is standard for older patients to get double plating and be full WB.
Conclusion - shorten and talk of double plate as a useful construct especially in the old. Do we need to plate the young? What about a RCT to answer this question?
References - Need 2 additions.
1) R Leighton et al JBJS (B) 2011 93B supp 3 305- LISS vs DCS in RCT
2) R Buckley et all Injury 42 194-199 2011 Malrotation in LISS implants.
Author Response
Reviewer #2:Generally this is an older technique. Surgeons have been using allograft struts for decades. This is never acknowledged by the authors. It should be. Many things about this paper need rewriting. See below.
Indeed, this is not a new technique and we acknowledged it. Cortical bone plate allograft was used in the treatment of periprosthetic fractures of the femur (Font-Vizcarra L, Fernandez-Valencia JA, Gallart X, Segur JM, Prat S, Riba J. Cortical strut allograft as an adjunct to plate fixation for periprosthetic fractures of the femur. Hip Int. 2010 Jan-Mar;20(1):43-9.) and distal femoral nonunion (Wang JW, Weng LH. Treatment of distal femoral nonunion with internal fixation, cortical allograft struts, and autogenous bone-grafting. J Bone Joint Surg Am. 2003 Mar;85(3):436-40.). However, there are not many relevant reports about the use of cortical bone plate allograft in the treatment of comminuted fractures of distal femur. Thus, for the patients with comminuted distal femoral fractures requiring medial auxiliary fixation, our treatment provides a choice.
- Why did it take 14 years to publish as the technology has improved a lot over the last 5 years? Why combine the older patients with the young as the young do not generally need medial fixation like the older patients. Also, older patients need to be full wt bearing immediately and not non wt bearing as mentioned. This needs to be rewritten along 2023 standards.
You raise a good question. Indeed, we expected to publish our article at the beginning of 2020. Due to the Covid-19 pandemics in China, we had to postpone publishing the data. The cortical bone plate allograft used in the series cases was provided by Xin Kang Chen Medical Technology Development Co., Ltd, Beijing, China. Unfortunately, the purchase contract between our hospital and the Xin Kang Chen Medical Technology Development Co., Ltd expired in 2015, and there was no renewal of the contract. Therefore, we had to stop enrolling cases in 2014.
The indications of a medial plate in addition to the lateral plate are medial supracondylar bone loss, low trans-condylar bicondylar fractures, medial Hoffa fracture, peri-prosthetic distal femur fractures, non-union after failed fixation with single lateral plate, poor bone quality and comminuted distal femur fractures (AO type C3). (Sain A, Sharma V, Farooque K, V M, Pattabiraman K. Dual Plating of the Distal Femur: Indications and Surgical Techniques. Cureus. 2019 Dec 27;11(12):e6483.) In this paper (Dual Plating of the Distal Femur: Indications and Surgical Techniques), the authors demonstrated two young patients treated with a medial plate in addition to lateral plating. Thus we believe that the use of double plate depends on the fractures type, independent of age.
- Introduction - Line 42-43 - This sentence is not needed - adds nothing.
We have deleted this sentence, please see the revised manuscript.
- - Line 50 - Full WB now standard for elderly patients after fixation of these fractures. Please change this writing as old ideas do not make for good publishing.
Thanks for your constructive suggestion. We have corrected it and please see the revised manuscript.
- - Line 53 - This is not a novel idea as allografts have been used with cerclage wires for peri-prosthetic fractures for decades. Please acknowledge and change this paragraph.
Sorry for the confusing. We acknowledge and change this paragraph.
- Patients and Methods - Why lump young and old patients together? They are different situations entirely. This reviewer recommends eliminating the few older patients and keeping the young patients (less than 60years) and making this a high energy paper. Most young people do not need a graft to heal with good technique. How do you justify using it in this young population?
Thanks for your constructive suggestion. First of all, this study was a retrospective analysis of existing clinical cases including young and old patients. Secondly, we believe that the use of a supplemental medial plate or cortical bone plate allograft depends on the fractures type of, independent of age. In this article (Bai Z, Gao S, Hu Z, Liang A. Comparison of Clinical Efficacy of Lateral and Lateral and Medial Double-plating Fixation of Distal Femoral Fractures. Sci Rep. 2018;8(1):4863.), the authors compared the clinical efficacy of lateral plate and lateral and medial double-plating fixation of distal femoral fractures and explore the indication of lateral and medial double-plating fixation of the distal femoral fractures. Most of the patients are young population.
- Line 64-83 - Most of this writing is "Results" - move to the next section. Why do you have 14 year old data? Why publishing so late when it is out of date? Much has changed over the last 14 years.
We have move Line 64-83 to the next section.
Due to the Covid-19 pandemics in China, we had to postpone publishing the data. The cortical bone plate allograft used in the series cases was provided by Xin Kang Chen Medical Technology Development Co., Ltd, Beijing, China. Unfortunately, the purchase contract between our hospital and the Xin Kang Chen Medical Technology Development Co., Ltd expired in 2015, and there was no renewal of the contract. Therefore, we had to stop enrolling cases in 2014.
7.Line 120 - Was there an incision proximally too? Please explain how screws were implanted proximally.
There are also an incisions proximally, and screws are inserted through minimally invasive techniques.
8.Line 129 - Were the LISS and the allograft connected by fixation or independent?
Sorry for the confusing.The cortical bone plate allograft were fixed in place with cortical bone screws in the LISS plate.
9.Discussion - Line 231 - You talk of periprosthetic fractures - were there some in your patient list?
Sorry for the confusing. Indeed, we talked of periprosthetic fractures, however, we just described the cases in the literature, there were no periprosthetic fractures in our patient list.
10.The Strengths and Limitations are mentioned very superficially in the Conclusions. Move to their own paragraph in the Discussion. Make more clear as there are problems such as 1) old data - 14 years old and much has changed in this time period. 2) Combining young and old population, 3) weight bearing was non WB when it is standard for older patients to get double plating and be full WB.
We have corrected it according to your advice and please see the revised manuscript.
11.Conclusion - shorten and talk of double plate as a useful construct especially in the old. Do we need to plate the young? What about a RCT to answer this question?
You raise a good question. A biomechanical study demonstrated that additional fixation of medial plate significantly increased the fracture stability in distal femur fractures fixed with the lateral locked plating. Especially in the clinical situations where sufficient stability cannot be provided at the distal segment, the medial plate may be considered as a useful biomechanical solution to obtain adequate stability for fracture healing (Park KH, Oh CW, Park IH, Kim JW, Lee JH, Kim HJ. Additional fixation of medial plate over the unstable lateral locked plating of distal femur fractures: A biomechanical study. Injury. 2019 Oct;50(10):1593-1598). Moreover, the indications of a medial plate in addition to the lateral plate are medial supracondylar bone loss, low trans-condylar bicondylar fractures, medial Hoffa fracture, peri-prosthetic distal femur fractures, non-union after failed fixation with single lateral plate, poor bone quality and comminuted distal femur fractures (AO type C3). (Sain A, Sharma V, Farooque K, V M, Pattabiraman K. Dual Plating of the Distal Femur: Indications and Surgical Techniques. Cureus. 2019 Dec 27;11(12):e6483.) In this paper (Dual Plating of the Distal Femur: Indications and Surgical Techniques), the authors demonstrated two young patients treated with a medial plate in addition to lateral plating. Thus we believe that the use of double plate depends on the fractures type, independent of age.
A long-term RCT study with a larger number of patients and control groups that include other fixation methods should be performed to further validate our findings here.
12.References - Need 2 additions.
1) R Leighton et al JBJS (B) 2011 93B supp 3 305- LISS vs DCS in RCT
2) R Buckley et all Injury 42 194-199 2011 Malrotation in LISS implants.
We have read the two references carefully and add them in the revised manuscript.

Round 2
Reviewer 1 Report
I congratulate you on thoroughly solving all the requests of the reviewers. The current form of the manuscript is much improved. Hope to see a future study of this type on a bigger number of patients.
Author Response
I congratulate you on thoroughly solving all the requests of the reviewers. The current form of the manuscript is much improved. Hope to see a future study of this type on a bigger number of patients.
We thank you for the insightful comments, which were not only scientifically meritorious but also extremely helpful in directing our efforts to enhance the scientific quality of this manuscript.
Reviewer 2 Report
Good revision but still some things to improve.
1) Methods - Line 142 - Was weightbearing really started immediately after surgery or have you just said so? You have changed the study with your writing. You initially said that you waited until callous formation. which is it? Be honest. Write it up as you did it. This reviewer will not accept it until the truth is known. Then add the problems with the study to the Limitations section in the Discussion which still needs to be constructed.
2) Conclusion - Line 312-316 - The authors still have not constructed a paragraph in the Discussion section which lists the strengths and limitations of the study. The Conclusion is too long. Take Line 312-316 out of the Conclusion and make an end paragraph in the Discussion section with all of the weaknesses like the study is small, retrospective with OLD data. Add this to the Limitations and then come up with a few strengths as your study has a few. Make the Conclusion shorter and do not repeat yourself.
Author Response
Good revision but still some things to improve.
We thank you for the insightful comments, which were not only scientifically meritorious but also extremely helpful in directing our efforts to enhance the scientific quality of this manuscript.
1)Methods - Line 142 - Was weightbearing really started immediately after surgery or have you just said so? You have changed the study with your writing. You initially said that you waited until callous formation. which is it? Be honest. Write it up as you did it. This reviewer will not accept it until the truth is known. Then add the problems with the study to the Limitations section in the Discussion which still needs to be constructed.
Sorry for the confusion. Indeed we made a mistake here and have corrected it. Please see the revised manuscript.
2) Conclusion - Line 312-316 - The authors still have not constructed a paragraph in the Discussion section which lists the strengths and limitations of the study. The Conclusion is too long. Take Line 312-316 out of the Conclusion and make an end paragraph in the Discussion section with all of the weaknesses like the study is small, retrospective with OLD data. Add this to the Limitations and then come up with a few strengths as your study has a few. Make the Conclusion shorter and do not repeat yourself.
Thanks for your constructive suggestion. We have corrected it and please see the revised manuscript.